# The Impact of Credit Constraints on International Quality and Environmental Certifications: Evidence from Survey Data

**Filomena Pietrovito** 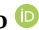

Department of Economics, University of Molise, 86100 Campobasso, Italy; filomena.pietrovito@unimol.it;
Tel.: +39-0874-404192

**Abstract:** Environmental and quality management practices are extremely relevant for a firm's development and international recognition. However, dealing with the standards required and obtaining an international standards certification involves costs for employee training, procedure documents, and third-party audit fees that must be paid in advance by companies. This paper attempts to analyze the impact of difficulties in accessing external financing on the likelihood of possessing the standards and the certification, for firms based in 64 countries. A crucial aspect in uncovering such a causal effect is the potential endogeneity of financial constraints with respect to international standards certification. I address this issue by adopting a bivariate probit model and a set of firm-level instrumental variables. The empirical results showed that financially constrained firms were less likely to possess the standards required and the associated international certification, and that this impact was more relevant for small and young firms.

**Keywords:** quality certification; credit rationing; survey data

## 1. Introduction

Internationally recognized standards certifications are an important step in the firm's development and international recognition. The most widely used standards certifications are the ISO 9000 series—for quality management systems—and the ISO 14000 series—for environmental management. Both certifications are issued by a non-governmental organization, the International Organization for Standardization (ISO), which verifies the conformity to the standards required. The ISO 9000 family, first published in 1987, is a set of managerial guidelines thus far applicable to any organization, regardless of its size, sector of activity, or business performance. The ISO 9000 family and its related standards are based on seven quality management principles, including customer focus—which requires, for instance, to align organizational objectives with customer needs and expectations and to measure customer satisfaction—and relationship management—which requires one to learn more about supplier quality and see resources related to managing the supply chain.[1] On the other hand, the ISO 14000 was established in 1996, within the framework of environmental management systems, as a voluntary instrument that helps organization to improve their environmental practices. The aim of the ISO 14000 is to encourage firm's incentives of being "green" and to reduce pollution (Bellesi et al. 2005).

In this framework, one relevant issue is that dealing with and implementing the standards imposed by guidelines require certain investments by companies, that often lack skills and management systems (Chen et al. 2019). Training of human resources, research and development, as well as procedure

---

[1] A complete list of principles is provided by ISO (2015).



updates are some examples of such costs. In addition, the adoption of ISO certifications involves costs associated with the bureaucracy, such as those of designing and reviewing documents, and certification bodies fees (Cagnazzo et al. 2010; Tian and Lin 2019).[2] Therefore, analyzing firm-specific factors that might prevent it from investing in quality management and environmental practices and dealing with the bureaucracy to obtain a certification is a relevant topic.

As financial support is crucial for such operations, this study sought to investigate across firms whether and how the ability to have access to external finance is related to the likelihood of possessing an internationally-recognized standards certification for quality management (such as, ISO 9000, 9002) and for environmental management (such as, ISO 14000), documenting the compliance of standards. To this end, I adopted firm-level data from the World Bank Enterprise Survey (WBES) for 64 countries, over the rounds 2006–2014.

The relevance of this research question relies on the characteristics of the costs detailed above, which a firm faces when it engages in international standards. These costs, in general, must be paid upfront, with the implication that the firms adopting internationally-accepted standards tend to demand external funds to a greater extent than firms that are not involved in international standards. Financial difficulties are magnified for small and medium-sized firms or for young firms, especially if based in less-developed countries, where capital markets and financial systems are often underdeveloped. An interesting issue arising in analyzing this relationship is that a firm might not acquire a certification because of two reasons—(i) it does not achieve certifiable standards, and therefore, the agency refuses to provide the certification, or (ii) it is discouraged from applying for a certification, even if it meets certain requirements in terms of quality and environmental issues, because of too complex procedures and documents or too high fees for the agency. On the other hand, if a firm possesses a certification, it is because it meets the requirements. As I will explain in Section 2, what I observe in our WBES data are whether a firms possesses a certification or not and, therefore, I measured both the likelihood of investing in quality and environmental management systems and the likelihood of dealing with documents and fees.[3] Moreover, in our data I are not able to distinguish between the agency-reasons for not acquiring a certification and the self-exclusion motivations for not applying for a certification.

Lots of attention in the literature is devoted to analyzing the effects of the compliance of environmental and quality standards from a microeconomic perspective, i.e., the level at which investments are conducted. From this point of view, the literature recognizes two categories of effects induced by international certifications—internal and external benefits (Chen et al. 2019). Internal benefits are related to increasing a firm's performances. Chen et al. (2019), for a sample of Chinese firms, and Ali and Yusuf (2019), for Indian firms, documented a beneficial impact of ISO certification on sales growth. It depended on the fact that, when adopting ISO standards, the firm is required to transform its production processes as well as quality of products and services. This provides a competitive advantage, both in domestic and in foreign markets, with the result of attracting more customers. Moreover, Ali and Yusuf (2019) find that certified companies create more job opportunities than the uncertified enterprises. A second beneficial impact is related to efficiency and productivity gains. Goedhuys and Sleuwaegen (2013) and Ullah et al. (2014) argue that productivity improvements might be associated with the fact that firms implement internationally accepted standards to improve the efficiency of the production process, as well as the quality of products and services. An additional source of efficiency gains comes from the reduction of fixed and variable transaction costs, between incompletely informed parties, obtained by respecting internationally agreed norms in the production and delivery process of products with a foreign destination. Finally, from a financial perspective,

---

[2]     According to Cao and Prakash (2011), the cost of a certification audit might fluctuate from 10,000 dollars to even 1 million dollars, depending on the companies' characteristics.

[3]     This is a common issue in the literature on certifications, as also recognized by Tian and Lin (2019). They assess that "a firm possessing an ISO 14000 certification means that it takes actions on environmental management. Thus, it is more likely to have better environmental performance than firms which don't acquire the certification" (Tian and Lin 2019, p. 433).

meeting international standards help reduce information asymmetries between creditors and the firm, when it needs to access the external capital market. This helps to increase the likelihood of obtaining external finance, as well as better financing conditions, since the certification signals that firms possess some characteristics related to its financial risk (Bellesi et al. 2005; Ullah et al. 2014; Mijatovic et al. 2019; Wu et al. 2020). In comparing developed and developing countries, Goedhuys and Sleuwaegen (2013) find that the largest gains in performances come from productivity improvements, helping firms in the developing markets to close the technological gap with developed countries.

On the other hand, external benefits are related to signals from the firm that help to reduce information asymmetries and improve its competitive position in the market in which it operates (Wu et al. 2020). Mijatovic et al. (2019) argue that certification is a sort of means of communication along the supply chain that ensures the buyer that the supplier meets certain requirements. In developing countries, it is able to support companies to overcome reputation problems and to enter foreign markets. International transaction benefits coming from international certifications affect domestic firms, in particular SMEs, that are required to meet productive requirements to enter new competitive markets (Goedhuys and Sleuwaegen 2016; Wu et al. 2020). Due to intense global competition, such incentives are likely to be higher for a developing country, where information asymmetries and reputational problems are critical among firms. In this case, buyers associate the quality of the seller with the generally poor reputation of the country where it is located (Potoski and Prakash 2009; Otsuki 2011; Bangwayo-Skeete and Moore 2015). Encouraging companies to increase transparency regarding their sustainable practices to different stakeholders, including government authorities, investors and insurance companies, is therefore crucial for their trust (Castka and Prajogo 2013; Ikram et al. 2020). Similarly, the certification procedures required by the ISO 9000 standards were developed with the aim to promote and make improvements in the global exchange of goods and services, especially in developing countries, building transparency in the quality management system (Ikram et al. 2020). Clougherty and Grajek (2008) find, for instance, that the adoption of ISO 9000 standards positively influences both arm's length international trade and foreign direct investments. While this result is interesting, they also find that ISO 14001 certification contributes more to the economic development than ISO 9001 and Social Accountability 8000 standards, in both developed and developing countries, with developing countries showing superior performances.

On the contrary, the effects of financial constraints on the likelihood to comply with standards and to obtain quality certification did not receive much attention in the existing literature. To the best of our knowledge, only Tian and Lin (2019) deal with the effects of financial constraints on the likelihood of obtaining environmental certification. Using a sample of 1547 Chinese firms retrieved from the WBES from 2011 to 2013, they showed that low credit constraints did not affect the environmental performance of companies, while moderate and severe financial obstacles negatively affected the firms' environmental performance. The idea was that when companies faced greater financial constraints, it was more challenging to address environmental issues, and therefore, to obtain certification. The existing empirical evidence on the relationship between credit constraints and internationally-accepted certifications was thus weak, needing additional evidence. This was surprising, provided the relevance of financial support and the beneficial economic impact of the certifications detailed above.

Our contribution to the literature is fourfold. First, to the best of our knowledge, this study is the first to examine the impact of credit rationing on quality standards certifications in a multi-country setting, provided that the scarce literature on this issue exploring single-country analysis. Fortunately, I can rely on an influential source of information, the WBES, that provides harmonized firm-level data comparable across a large sample of countries, containing not only registry information, but also firm performances. Second, I adopt a measure of financial constraints based on answers to two questions included in the WBES. The first question was whether the firm was rejected for a loan by a bank or a financial intermediary. The second question, for every firm that needed external finance but did not apply for a loan, investigated the main reason that discouraged it from applying for a loan. Third,

our estimation methodology on firm-level data relied on an instrumental variable approach to tackle the endogeneity problems that might otherwise affect our empirical relationship. This enabled us to better identify causal effects and establish more directly the impact of financial constraints on ISO certification.[4] Fourth, I exploited the firm-level heterogeneity by testing the impact of credit rationing on different sub-samples of firms, depending on size and age.

The empirical results showed that financially constrained firms were about 10 per cent less likely to obtain an international standards certification than unconstrained firms. This impact was more relevant for the sub-sample of small firms (with less than 24 employees) and of young firms (those with less than 16 years of activity).

This paper is structured as follows. Section 2 describes the data used and its source. Section 3 presents the empirical model adopted to test the relationship between credit constraints and international standards certification and to address the endogeneity issue. Section 4 discusses the baseline results, also distinguishing between small and medium-large firms and young and more experienced firms. Section 5 concludes.

## 2. Data and Sources

To test the impact of credit rationing on the likelihood of acquiring an international standards certification, I adopted a database that was also used in Pietrovito and Pozzolo (2019), and complemented with information on ISO certifications. Data were retrieved from the WBES, an influential source of harmonized firm-level data comparable across a large sample of countries, containing not only registry information (such as ownership, sector of economic activity and size), but also firm performances. The WBES is extensively used in the empirical literature and presents data in a variety of ways, including the panel structure.[5] The original panel dataset retrieved from the WBES included 111,425 firms (and 117,095 observations) ranging across all economic activities and spanning 73 countries, mostly developing ones, over the waves of 2003 and of the period 2006–2014. Starting with all observations, I restricted our focus on the manufacturing firms and excluded firms for which the variables adopted in the empirical analysis exhibited missing values. In this way, I ended up with a sample of 25,034 observations for 24,763 private firms belonging to 19 manufacturing industries—defined according to the ISIC at 2 digits level.[6] Since the panel component in this sample included only a limited number of firms (271), our preferred strategy was to present the results obtained from the pooled cross-section sample, over the period of 2006–2014. Table S1 in the Supplementary Materials provides the sample composition by country, the relative share of certified firms and years surveyed, showing that 14 per cent of countries had one survey, 70 per cent had two surveys, and 16 per cent surveyed for three years.

Of particular interest to our research was that this database contained information about whether the company possessed internationally recognized quality certifications (such as ISO 9000, 9002, and 14000). The relative question in the survey was 'Does the organization have an internationally recognized quality certification?' The possible answers were: 'Yes', 'No', 'Still in progress', and 'Don't know.' Excluding firms with the last two answers, I construct a dichotomous variable (ISO certification) assuming the value of 1 for firms that acquired an accepted international standards certification and zero otherwise. In our empirical model, this variable was used as the dependent variable. As I pointed out in Section 1, this dichotomous variable captured both the ability of the firm to invest in quality and

---

[4]  The measure of credit constraints as well as the methodology adopted in this paper mirror those developed in Pietrovito and Pozzolo (2019) and Nucci et al. (2020) in the literature on credit constraints and international trade.

[5]  Firms included in an Enterprise Survey are extracted from stratified samples depending on size, sector of economic activity and geographic region within a country. Data are publicly accessible at http://www.enterprisesurveys.org.

[6]  In the sample used for the empirical analysis and in order to avoid inflating the standard errors by repeated observations, the panel component includes only firms that have changed either their ISO certification status or their status of being credit rationing. I believe that this strategy is appropriate since the time span between a firm's survey and the next one is quite large.

environmental practices and the recognition obtained by the ISO for this effort.[7] Table 1 presents the descriptive statistics and shows that about 25 per cent of our sample of firms was ISO-certified, in line with other studies on a similar sample of countries (see, for example, Ullah et al. 2014).

Regarding our focal independent variables, empirical research estimated the difficulty of entering the credit market through the following information—(i) the company's balance sheet and cash flow statement items (leverage and liquidity ratios), (ii) credit scoring, containing comprehensive information about the company's risks, and (iii) the company's self-assessment of the difficulty of obtaining financing from external sources (Ullah et al. 2014; Tian and Lin 2019).

When constructing credit rationing indicators, I followed the method first proposed by Jappelli (1990) and adopted by Pietrovito and Pozzolo (2019) and Nucci et al. (2020) in the literature on credit constraints and international trade. To construct this variable, I relied on the availability of detailed motivations provided by the respondents in the WBES. In particular, I adopted two main reasons to identify a credit rationed firm that needed a loan. The first reason depended on banking decision and allowed to define credit rationed firms as those that applied for a loan but did not obtain it through the financial intermediary. The second reason depended on self-exclusion motivations, such as loan conditions—in terms of contract length, interest rates, collaterals, and procedures—and pessimistic expectations that the loan would be rejected.

From the descriptive statistics reported in Table 1, I found that on average 21 per cent of firms exhibited some financial constraints and that this share was higher for firms that did not have a quality certification (23 per cent) than for ISO-certified firms (13 per cent). In general, these descriptive statistics, and the associated *t*-test, suggest that firms with a quality certification show a significant lower difficulty in accessing credit, preliminarily verifying our main hypothesis that financial constraints can effectively reduce the company's certification.

To reduce the risk of bias arising from the omitted relevant variables in the empirical specification, the information on quality certification and credit rationing is supplemented by that on other individual characteristics (see, for example, Ullah et al. 2014; Tian and Lin 2019). First, the likelihood of a quality certification might be affected by the structural characteristics, which were captured in our analysis through—(i) the number of permanent full-time employees (size), (ii) labour productivity, measured by the ratio of total sales to the number of employees (labor productivity), and (iii) firm age (age), measured by the number of years since its foundation. The empirical literature also extensively considered the complementarity between foreign market orientation and quality certification (see, for example, Pekovic 2010). For this reason, I additionally controlled for a dummy variable that reflected the firm's exporting status (export) and took the value of one if the firm exported its products to foreign markets, and zero otherwise. Additionally, to account for the legal structure, I controlled for foreign ownership, that is, the percentage of the firm that was owned by private foreign individuals, companies or organizations. I expected that both export-oriented and foreign owned firms have a higher opportunity to deal with standards and to obtain internationally-recognized certifications, because of their success in affording the related extra costs (Tian and Lin 2019).

---

[7] One can raise the issue that, while there might be some correlation between different types of certifications, they are not substitutable. While this is the case in this paper, since the dummy lumps from certifications about management standards to environmental issues, the current design of the WBES precludes identification of different types of certifications. I would thank an anonymous reviewer for pointing out this issue.

**Table 1.** Descriptive statistics.

| Variable | (1) All Sample | | | | (2) Non-Certified | | | | (3) ISO-Certified | | | | (4) *t*-Test | |
|---|---|---|---|---|---|---|---|---|---|---|---|---|---|---|
| | Mean | c.v. | Min | Max | Mean | c.v. | Min | Max | Mean | c.v. | Min | Max | | |
| ISO certification | 0.247 | 1.746 | 0 | 1 | - | - | - | - | - | - | - | - | | |
| CR | 0.209 | 1.948 | 0 | 1 | 0.234 | 1.810 | 0 | 1 | 0.132 | 2.565 | 0 | 1 | 19.23 | *** |
| Employees | 109.733 | 3.828 | 0 | 26,000 | 59.403 | 3.196 | 0 | 11,500 | 263.182 | 2.877 | 0 | 26,000 | −20.95 | *** |
| Labour productivity | 36,954 | 1.566 | 0 | 440,961 | 29,020 | 1.609 | 0 | 440,625 | 61,145 | 1.280 | 0 | 440,961 | −30.54 | *** |
| Age | 21.522 | 0.821 | 1 | 202 | 20.098 | 0.810 | 1 | 202 | 25.862 | 0.802 | 1 | 166 | −19.92 | *** |
| Export status | 0.342 | 1.388 | 0 | 1 | 0.251 | 1.730 | 0 | 1 | 0.620 | 0.783 | 0 | 1 | −53.30 | *** |
| Foreign ownership | 0.070 | 3.393 | 0 | 1 | 0.046 | 4.234 | 0 | 1 | 0.142 | 2.271 | 0 | 1 | −22.04 | *** |
| Female ownership | 0.328 | 1.432 | 0 | 1 | 0.323 | 1.449 | 0 | 1 | 0.343 | 1.384 | 0 | 1 | −2.94 | *** |
| Savings accounts | 0.861 | 0.402 | 0 | 1 | 0.835 | 0.445 | 0 | 1 | 0.940 | 0.253 | 0 | 1 | −25.93 | *** |

Notes: Panel (1) reports the descriptive statistics calculated on the whole sample. Panels (2) and (3) report the descriptive statistics calculated on the sub-samples of ISO-certified and non-certified firms, respectively. Panel (4) reports the value of the mean-difference test. *** denote significance at 1 per cent level.

**Table 2.** Correlations.

| | | ISO Certification | CR | Employees | Labour Productivity | Age | Export Status | Foreign Ownership | Female Ownership | Savings Accounts |
|---|---|---|---|---|---|---|---|---|---|---|
| (1) | ISO certification | 1 | | | | | | | | |
| (2) | CR | −0.108 * | 1 | | | | | | | |
| (3) | Employees | 0.209 * | −0.068 * | 1 | | | | | | |
| (4) | Labour productivity | 0.239 * | −0.105 * | 0.078 * | 1 | | | | | |
| (5) | Age | 0.141 * | −0.062 * | 0.184 * | 0.124 * | 1 | | | | |
| (6) | Export status | 0.336 * | −0.119 * | 0.187 * | 0.205 * | 0.140 * | 1 | | | |
| (7) | Foreing ownership | 0.174 * | −0.027 * | 0.117 * | 0.154 * | 0.031 * | 0.189 * | 1 | | |
| (8) | Female ownership | 0.019* | −0.054 | 0.011 * | −0.013 * | 0.047 * | 0.045 * | −0.060 * | 1 | |
| (9) | Savings accounts | 0.131 * | −0.122 * | 0.061 * | 0.109 * | 0.061 * | 0.146 * | 0.063 * | 0.072 * | 1 |

Notes: * denotes significance at 5 per cent level.

Considering the individual firm-level characteristics detailed above, descriptive statistics in Table 1 indicate that the average number of employees was 110, with ISO-certified firms being larger than the non-certified firms (263 vs. 59 permanent employees). Moreover, considering other structural characteristics, I find that firms without an ISO certification were also less productive and younger than firms with a certification, with values of 29,020 vs. 61,145 dollars and 20 vs. 26 years, respectively. Additionally, Table 1 reports, for non-certified firms, a significantly lower probability of exporting (25 per cent) than firms with a quality certification (62 per cent) and a significantly lower share of foreign ownership (5 per cent) than certified firms (14 per cent). These results confirmed that exporting and foreign owned firms were more likely to afford the sunk extra costs related to the procedure and management required to obtain quality certification.

I additionally addressed the issue that the relationship between credit rationing and firms' quality certification might suffer from at least two major endogeneity problems. First, unobserved firm-level characteristics might influence both their ability to access external finance and whether or not they possess ISO-certifications. Second, the relationship under scrutiny might be due to reverse causation, since an internationally-recognized quality certification might be seen as a positive signal that makes it easier to obtain external funding, reducing the problems of credit constraints. Several researchers showed that when a company has a good reputation and complete information disclosure, it is more likely to obtain external financing and better financing conditions (Ullah et al. 2014; Wu et al. 2020).

The instrumental variables adopted to address the endogeneity issue were selected within the strand of the literature investigating the firm-level attributes that might signal the ability of a borrower to repay a loan and reduce the informational asymmetries that typically characterize a bank-firm relationship. Studies in this strand of the literature argued that lenders must resort to some borrower's characteristics, in order to decide who obtains a loan and its amount (Stiglitz and Weiss 1981; Diamond 1991). In a recent contribution, Drakos and Giannakopoulos (2011) showed that the probability of credit rationing depended on both the fact that the principal owner of the firm was a female and on maintenance of a savings account, among other determinants. In Drakos and Giannakopoulos (2011), gender differences took the form of a high probability of rationing when the principal owner of the firm was female, implying some correlation between credit constraints and risk aversion. Moreover, they found that firms maintaining savings accounts were more likely use credit rationing.

Therefore, I adopted two instruments for credit rationing that are available from the current design of the WBES. The first was a dummy indicating whether there were any females amongst the owner of the firm (Female ownership) and captured the experience of females in business management. The second instrumental variable, Savings accounts, was a dummy equal to one for firms with a checking or a savings account and reflected the relationship with the bank and the previous use of a lender. One concern with our instruments was that they might still be correlated with some omitted variables driving both the probability that a firm was credit rationed and its internationally recognized certifications. This was the case if, for instance, I were not adequately measuring our control variables and their unmeasured components were correlated with the instruments. Reassuringly, in all our specifications, the Hansen test provided evidence of the exogeneity of our instruments.

From Table 1, it could be inferred that about 33 per cent of our sample had females among its owners and about 86 per cent of our firms had a checking or a savings account.

To verify the multicollinearity, Table 2 presents the correlations between variables at the 5 per cent significance level. The correlations of the variable indicating whether the firm was ISO certified and our measure of financial constraints was negative (−0.108) and significant at the 5 per cent level. On the contrary, ISO certification was positively correlated with the control variables—size (0.209), productivity (0.239), age (0.141), export status (0.336), and foreign ownership (0.174).

## 3. The Empirical Approach

To analyze the impact of credit constraints on the likelihood of possessing an internationally -recognized standards certification, I adopted the following specification, where $i$ is the index for firm, $k$ represents the sector of economic activity, $c$ represents the country and $t$ represents time:[8]

$$Pr(ISO\ certification_{ikct} = 1) = Pr(\alpha + \beta CR_{ikct} + \gamma Z_{ikct} + \nu_k + \lambda_c + \eta_t + \varepsilon_{ikct} > 0 = \\ \Phi(\beta CR_{ikct} + \gamma Z_{ikct} + \nu_k + \lambda_c + \eta_t) \tag{1}$$

The dependent variable, *ISO certification$_{ikct}$*, was equal to one if the firm had an internationally-recognized quality certification at time $t$ and zero otherwise. $CR_{ikct}$ is a binary variable equal to one if firm $i$ experienced financial difficulties at time $t$ and zero otherwise. $Z_{ikct}$ is the vector of control variables outlined in the previous section, including size, productivity, age, export status and foreign ownership. In the empirical specification, size, productivity and age were in logarithm. Besides the firm-level characteristics, I also controlled for three sets of fixed effects that represent additional factors affecting the probability of obtaining an international certification. The first set was $\nu_k$, which reflected time-invariant, sector-specific effects. The second set was $\lambda_c$, which reflected the time-invariant, country-level effects that might impact on certification, such as institutional environment and regulations, as well as financial and human resource constraints that might hinder the general market functioning (Massoud et al. 2010; Ullah et al. 2014; Goedhuys and Sleuwaegen 2016). Moreover, country fixed-effects might reflect the fact that firms in countries with poor reputation would have more incentive to go out of their way to get internationally certified, in order to gain access to external markets relative to the firms located in markets that would not make them look as low quality (Potoski and Prakash 2009; Otsuki 2011; Bangwayo-Skeete and Moore 2015). The third was $\eta_t$, which reflected any time-specific shocks simultaneously affecting all countries. $\varepsilon_{ikct}$ is a normally distributed random error with a zero mean and unit variance.

Equation (1) was estimated by adopting two estimators—the linear probability model (LPM) and the binary choice probit model. While on the one hand, they allowed the estimation of the predicted probability of possessing an ISO certification, controlling for other firm-level characteristics and for fixed effects, they did not account for the potential endogeneity of credit rationing with respect to the likelihood of acquiring an internationally-accepted certification. Moreover, the LPM did not take into account the fact that the dependent variable was dichotomous. For this reason, I additionally estimated our model with an instrumental variable methodology, that also took into account the fact that both ISO certification—our dependent variable—and CR—our main regressor—were dichotomous. I thus estimated with a maximum-likelihood, a two-equation probit model (a bivariate probit, bi-probit) in which the first-level equation was the following:

$$Pr(CR_{ikct} = 1) = Pr(\delta I_{ikct} + \lambda Z_{ikct} + \psi_k + \tau_c + \varsigma_t + \mu_{ikct} > 0) \\ = \Phi(\delta I_{ikct} + \lambda Z_{ikct} + \psi_k + \tau_c + \varsigma_t) \tag{2}$$

Equation (2) estimated the probability that a firm was in financial difficulties as a function of the set of instrumental variables, $I_{ikct}$, and the same set of exogenous variables included in Equation (1), $Z_{ikct}$. The second-level equation of the bi-probit model was identical to Equation (1) and estimated the probability of possessing an ISO certification, conditional on being credit rationed. Moreover, $\mu_{ikct}$ in Equation (2) was a normally distributed random error with zero mean and unit variance, which in a recursive bivariate probit model, was allowed to be correlated to the error term $\varepsilon_{ikct}$ in Equation (1). The potential endogeneity between credit rationing and the likelihood of being ISO certified arose because of the possible correlation between the unobserved determinants of credit

---

[8] The same methodology has been adopted by Pietrovito and Pozzolo (2019) and Nucci et al. (2020).

rationing, which were included in the error term of Equation (2), and the unobserved determinants of the ISO certification, included in the error term of Equation (1). As argued by Minetti and Zhu (2011) and Minetti et al. (2019), in presence of such correlation, the bi-probit model was the appropriate estimator due to two reasons—(i) it allowed us to take into account the correlation between $\mu_{ikct}$ and $\varepsilon_{ikct}$, and (ii) the impact of credit rationing on the likelihood of possessing an international certification was identified by the exclusion of the set of instrumental variables, $I_{ikct}$, from Equation (1).

## 4. Results

### 4.1. Baseline Results Controlling for Endogeneity

This section describes the results obtained by estimating Equation (1) with the different methodologies adopted. The baseline results are reported in Table 3 and are based on 25,034 observations. Column 1 reports the results obtained by estimating Equation (1) with LPM, where the dichotomous dependent variable was treated as a continuous one. As expected, the coefficient of CR was negative and equal to −0.010, statistically significant at the 10 per cent level. This indicated that credit constrained firms were 1 per cent less likely to obtain an internationally-recognized certification than not-credit constrained firms.

The coefficients of the control variables revealed that larger, more productive, and older firms were also more likely to possess an internationally-recognized certification, as shown by the estimated coefficients of (the logarithm of) the number of workers (0.090), the (logarithm of) labor productivity (0.024) and the (logarithm of) age (0.012). The dummy reflecting the exporter status also positively impacted on the likelihood of obtaining a quality certification, with a coefficient equal to 0.148, statistically significant at the 1 per cent level. This result indicated that exporting firms were 14.8 per cent more likely to obtain a certification that non-exporting firms. This impact was consistent with previous literature (e.g., Ullah et al. 2014). Similarly, the impact of foreign ownership was positive and highly statistically significant. Foreign owned firms were indeed 11 per cent more likely to obtain a quality certification than domestically owned firms.

Column 2 revealed that estimating a probit model provided very similar impacts compared to that obtained with the LPM. In particular, the marginal effect of credit rationing was −0.013, indicating that constrained firms were 1.3 per cent less likely to obtain a certification. The impact of control variables was comparable to that of the LPM, both in terms of marginal effect and of statistical significance.

As detailed above, in studying the impact of financial constraints on ISO certifications, one incurs in endogeneity issues. The bi-probit model was adequate to deal with this concern, jointly estimating two equations—the one for the probability that a firm was credit rationed (first-level equation) and the other for the probability that a firm possessed an ISO standards certification (second-level equation). The results, reported in Column 3 of Table 3, showed that the correlation between the error terms of the two equations, $\mu_{ikct}$ and $\varepsilon_{ikct}$, was positive and statistically significant at the 5 per cent level. This confirmed that the likelihood of being credit rationed and that of possessing an ISO certification were jointly determined, such that an unobserved shock that increased the probability that a firm was credit rationed also showd a positive and direct impact on the probability that the same firm obtained a quality certification. Reassuringly, the bi-probit specification confirmed the negative impact of credit constraints on ISO certification, even though the marginal effect was larger than that obtained in Columns 1 and 2. The value of −0.095, statistically significant at the 10 per cent level, pointed to a significant impact from an economic perspective. Credit constrained firms were 9.5 per cent less likely to possess a certification than firms not experiencing financial difficulties. As pointed out above, this reflected both the ability to comply with standards and the recognition obtained by the ISO for this investment. Reassuringly, the other estimated coefficients were broadly unchanged when I controlled for the endogeneity of credit rationing, suggesting that the unobserved determinants of being rationed

that were correlated with the condition of obtaining a certification were not significantly correlated with other firm characteristics.[9]

**Table 3.** Impact of credit rationing on ISO certification.

| Variables | (1) OLS | (2) Probit | (3) Bi-Probit |
|---|---|---|---|
| CR | −0.010 * | −0.013 * | −0.095 * |
|  | (0.006) | (0.008) | (0.053) |
| Employees (log) | 0.090 *** | 0.079 *** | 0.076 *** |
|  | (0.002) | (0.004) | (0.005) |
| Labour productivity (log) | 0.024 *** | 0.025 *** | 0.023 *** |
|  | (0.002) | (0.001) | (0.002) |
| Age (log) | 0.012 *** | 0.011 ** | 0.010 ** |
|  | (0.004) | (0.005) | (0.005) |
| Export status | 0.148 *** | 0.120 *** | 0.117 *** |
|  | (0.006) | (0.012) | (0.011) |
| Foreign ownership | 0.112 *** | 0.078 *** | 0.079 *** |
|  | (0.010) | (0.009) | (0.010) |
| *Instrumental variables* |  |  |  |
| Female ownership |  |  | −0.014 *** |
|  |  |  | (0.005) |
| Savings accounts |  |  | −0.042 *** |
|  |  |  | (0.011) |
| corr[$\varepsilon_{ikct}$, $\mu_{ikct}$] |  |  | 0.203 (0.029) |
| Kleibergen-Paap first stage *F*-statistic (*p*-value) |  |  | 14.90 (0.000) |
| Overidentifying restrictions statistic (*p*-value) |  |  | 2.366 (0.124) |
| *N* | 25,034 | 25,034 | 25,034 |

*Notes*: Panel 1 reports the coefficients of the LPM and Panels 2 and 3 report marginal effects of the probit and bivariate probit (bi-probit) models. In Panel 3, the measure of credit rationing is instrumented using a dummy variable indicating whether there are any females among the owners of the firm (Female owners) and a dummy equal to one for firms with a savings or a checking account (Savings account). Unreported fixed effects for sector, country, and year are included in all regressions. Robust standard errors are clustered by sectors and reported in parentheses; corr[$\varepsilon_{ikct}$, $\mu_{ikct}$] is the correlation coefficient ($\rho$) between the unobserved determinants of the ISO certification dummy ($\varepsilon_{ikct}$) and those of rationing ($\mu_{ikct}$). The Kleibergen-Paap first stage *F*-statistic (*p*-value) is the value of the *F* statistic (with the *p*-value) for the hypothesis that instruments have jointly zero coefficients in the first stage regression. The over-identifying restrictions statistic (*p*-value) is the value of the Hansen statistic (and *p*-value). Kleibergen-Paap first stage *F*-statistic (*p*-value) and overidentifying restrictions statistic (*p*-value) are obtained from the two-stage least-squares estimation of the companion specification for ISO certification, where credit rationing is instrumented using its instruments. ***, **, * denote significance at 1 per cent, 5 per cent, and 10 per cent levels.

Both instruments adopted in the first stage regression showed a negative and significant impact on the probability of credit rationing. Firms with female owners and those with a savings or checking account were, respectively, 1.4 per cent and 4.2 per cent less likely to be credit constrained than other firms. The validity of our instrumental variables was tested using the Kleibergen-Paap and the Hansen tests. As noticed by Mikusheva (2013), the literature on statistical inference on weak instruments for non-linear models was much less developed than that on linear IV models. For this reason, I estimated both statistics from the two-stage least-squares estimation of the companion specification for ISO certification (Equation (1)), where credit rationing was instrumented using its instruments.

---

[9] Since one can argue that credit rationing, the key explanatory variable in Equation (1), was estimated in Equation (2), the standard errors should be corrected for this. To this end, I re-estimated the bi-probit model by bootstrapping standard errors, with 100 replications, and the results (available on request) remain unchanged.

The Kleibergen-Paap tested the weak identification hypothesis, which arose when the excluded instrument were weakly correlated with the endogenous regressors and it rejected the null hypothesis with a *p*-value of 0.000, as reported at the bottom of Table 3. The Hansen test of the overidentifying restrictions tested the joint null hypothesis that the instruments were valid, i.e., uncorrelated with the error term and correctly excluded from the estimated Equation (1). The Hansen test statistic was distributed as chi-squared with a *p*-value of 0.124, as reported in Table 3. Therefore, I failed to reject the null hypothesis, a case suggesting that our instruments were valid.[10]

### 4.2. Firm-Level Characteristics

Up to now, the baseline results showed that the average estimated impact of credit constraints on the likelihood of obtaining an internationally-recognized certification was 9.5 per cent, controlling for endogeneity. However, this impact might differ depending on some characteristics of the firms. In particular, since our sample included about 25,000 firms, heterogeneous in terms of size and age, our interest now focused on re-estimating the baseline empirical model considering different sub-samples of firms.

As argued above, small firms were likely to experience stronger difficulties in anticipating the costs of managing a quality certification procedure than larger firms. As a result, the impact of credit-constraints on the likelihood of possessing an ISO certification of these firms might be stronger than that on larger ones. To this end, I distinguished two sub-samples of firms, adopting the definition of the WBES—small firms, those with less than 24 employees (the median level), medium and large firms with a number of employees higher than 24. The results are reported in Table 4. They document that for the sub-sample of 12,562 small firms, the impact of CR was negative and statistically significant at the 1 per cent level, and it was twice the average effect reported in Column 3 of Table 3. Financially constrained small firms were therefore 21 per cent less likely to obtain ISO certifications, compared to other small firms that were not financially constrained. On the contrary, the marginal effect of CR estimated on the sub-sample of medium and large firms (12,472 observations) was not statistically significant, even if it was still negative. This result indicated that, within this subsample, credit rationing was not as relevant as in the case of small firms. This empirical result was in line with the previous literature documenting that small firms experienced higher difficulties in general, both in accessing the financial markets and in dealing with quality and environmental standards.

Considering the second firm-level characteristic, I re-estimated the bi-probit model on the sub-sample of young firms, with less than 16 years of activity, and of more experienced firms, as those with more than 16 years of activity since their foundation. The results, in line with our expectations, are reported in Columns 3 and 4 of Table 4. For young firms, the marginal impact of credit rationing on ISO was larger than the average and was equal to 16 per cent, statistically significant at the 1 per cent level. Contrarily, the marginal impact estimated for the sub-sample of 12,288 experienced firms was still negative, but statistically non-significant.

The results discussed in this Section demonstrated that even though I found an average negative impact of difficulties in accessing the credit market on the likelihood of possessing an ISO certification, this impact seemed to affect more small and young firms, which were probably those experiencing higher financial constraints in developing countries.

---

[10] These tests are obtained using the Stata routine ivreg2.

**Table 4.** Sample split by firm-level characteristics.

| Variables | (1) | (2) | (3) | (4) |
|---|---|---|---|---|
| | Small | Medium-Large | Young | Experienced |
| CR | −0.208 *** | −0.019 | −0.161 *** | −0.046 |
| | (0.065) | (0.080) | (0.051) | (0.072) |
| Employees (log) | 0.047 *** | 0.103 *** | 0.067 *** | 0.084 *** |
| | (0.007) | (0.006) | (0.006) | (0.006) |
| Labour productivity (log) | 0.014 *** | 0.031 *** | 0.018 *** | 0.028 *** |
| | (0.002) | (0.003) | (0.002) | (0.003) |
| Age (log) | 0.007 | 0.012 * | 0.006 | 0.012 |
| | (0.004) | (0.007) | (0.008) | (0.008) |
| Export status | 0.082 *** | 0.155 *** | 0.098 *** | 0.134 *** |
| | (0.007) | (0.016) | (0.008) | (0.015) |
| Foreign ownership | 0.081 *** | 0.093 *** | 0.078 *** | 0.082 *** |
| | (0.018) | (0.015) | (0.015) | (0.016) |
| *Instrumental variables* | | | | |
| Female ownership | −0.014 ** | −0.012 ** | −0.023 ** | −0.005 |
| | (0.006) | (0.006) | (0.010) | (0.007) |
| Savings accounts | −0.062 *** | −0.035 *** | −0.047 *** | −0.036 ** |
| | (0.015) | (0.010) | (0.010) | (0.015) |
| Kleibergen-Paap first stage *F*-statistic (*p*-value) | 9.25 (0.002) | 19.10 (0.000) | 17.74 (0.000) | 4.68 (0.023) |
| Overidentifying restrictions statistic (*p*-value) | 0.096 (0.756) | 1.275 (0.260) | 1.796 (0.180) | 2.261 (0.132) |
| N | 12,562 | 12,472 | 12,746 | 12,288 |

Notes: The table reports the marginal effects of the bivariate probit (bi-probit) models estimated on different sub-sample of firms. Panels 1–2 report the results for small firms (with less than 24 employees) and medium and large firms (with more than 24 employees), respectively. Panels 3–4 report the results for young firms (with less than 16 years of experience) and experienced firms (with more than 16 years), respectively. Robust standard errors are clustered by sectors and reported in parentheses. The Kleibergen-Paap first stage *F*-statistic (*p*-value) is the value of the *F* statistic (with the *p*-value) for the hypothesis that instruments have jointly zero coefficients in the first stage regression. The over-identifying restrictions statistic (*p*-value) is the value of the Hansen statistic (and *p*-value). Kleibergen-Paap first stage *F*-statistic (*p*-value) and overidentifying restrictions statistic (*p*-value) are obtained from the two-stage least-squares estimation of the companion specification for ISO certification, where credit rationing is instrumented using its instruments. ***, **, * denote significance at 1 per cent, 5 per cent, and 10 per cent levels.

## 5. Concluding Remarks

In this study, I dealt with the issue of external financial constraints that might hinder a firm's performance in terms of internationally-recognized certifications. The issue of financial constraints is particularly relevant for small and young firms operating in developing countries. To this end, I adopted a large database including 24,763 firms operating in 64 countries, obtained from the World Bank Enterprise Survey (WBES)—an influential source of micro data comparable across countries.

To analyze the impact of financial constraints on the likelihood of acquiring an ISO certification, I adopted an econometric specification that took into account not only the fact that both our dependent variable and our main regressor were dichotomous, but also the potential endogeneity of credit constraints with respect to ISO certification. In particular, I adopted the bivariate probit model and firm-level instrumental variables that allow to deal with both issues.

The results of our study showed that credit constraints have a negative and significant impact on the likelihood of possessing an ISO certification. Since our dichotomous dependent variable captured both the ability of the firm to invest in quality and environmental practices and the recognition obtained by the ISO for this effort, this indicated that firms facing difficulties in accessing financial markets, because of intermediary reasons or because of self-exclusion, were less likely to meet certain

standard requirements and therefore to obtain a certification for this investment. The average impact in our sample was 9.5 per cent, which was also economically relevant compared to the probability of possessing an ISO certification equal to 25 per cent. Moreover, since firms in our sample were heterogeneous in terms of individual characteristics, I then re-estimated the bivariate probit model on different sub-sample of firms, depending on size and age. In this respect, I found that on the sub-sample of small firms—with less than 24 employees—and on the sub-sample of young firms—those with less than 16 years of activity since their foundation—this impact was higher than the average. In addition, on the other sub-samples of medium and large firms and of more experienced ones, the impact was still negative, but not significant.

Therefore, our results were consistent with the expectation that financial constraints might be an obstacle for firms to deal with quality and environmental standards, and with the documents and procedures required by international agencies in less-developed countries, and that these constraints are more relevant for firms whose access to financial markets are already prevented by their size and experience.

The policy implications of these results deal with the efforts that policy makers should realize to facilitate the access to external finance by firms, in order to stimulate their investments in quality and environmental practices and their ability to deal with documents and procedures required by the ISO certifications. These efforts would benefit more small and young firms that are in general more financially constrained than larger and more experienced firms.

**Supplementary Materials:** The following are available online at http://www.mdpi.com/1911-8074/13/12/322/s1, Table S1: Distribution of ISO-certified firms and survey years.

**Funding:** This research received no external funding.

**Acknowledgments:** I would like to thank Alberto Franco Pozzolo for his valuable comments on a previous version of the manuscript. I would also recognize the suggestions of Agapito Emanuele Santangelo.

**Conflicts of Interest:** The author declares no conflict of interest.

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
