# Peer review of "The Impact of Credit Constraints on International Quality and Environmental Certifications: Evidence from Survey Data"

_jrfm, doi:10.3390/jrfm13120322_

Round 1

Reviewer 1 Report

The author attempts to analyze the impact of lack of external financing access on the likelihood of the firms to possess international quality and environmental certifications. Using firms level data from 70 countries and pooled estimations the authors show that financially constrained firms are less likely to apply and get international certification.

My comments to the author are as follows:

 My main concern with the manuscript is that there are serious issues with the identification. Both instrumental variables are problematic, and I do not think that they resolve the endogeneity problem.

a. The first instrumental variable is a dummy variable that takes the value of one if the firm’s financial statement is checked and certified by an external auditor and zero otherwise (financial statement certification). One of the requirements for ISO certification is that firm’s financial statement is checked and certified by an external auditor. It makes not sense to use as an instrumental variable part of the requirements for the dependent variable. If anything, you are explaining dependent variable with part of the dependent variable itself.

b. The second instrument is a measure of availability of internal sources of funds, constructed as the proportion of total annual sales of firm’s products that are paid for after delivery (delayed payments). It is not clear what they author is trying to measure here. How does the IV impact only independent varible without impacting the dependent varible. The author does not give any explanation what is the role of this instrumental variable, and how it helps us explain the relationship between the bidirectional impact of credit constraints and certification of international standards.

Here are some other comments that the author can use while she revises the paper.

  1. The author needs to rethink the dependent variable. She needs to explain why she lumps in ISO 9000 and ISO 14000. One is about standards of management of the company, while the other one is about environmental considerations of the company. While there might be some correlation, they are not substitutable.
  2. The columns in Table 1 are mislabeled which gave me a hard time (and potentially many other readers in the future). ISO-certified versus non-certified columns should be swapped. As it stands it shows that non certified firms have more employees, are more productive, are older, export more, are more owned by foreigners, etc.
  3. The author might want to consider that firms in countries with poor reputation would have more incentive to go out of their way to get internationally certified to be able to have access to external markets relative to the firms located in markets that would not make them look as low quality. (Potoski and Prakash 2009; Otsuki 2011; Bangwayo-Skeete and Moore 2015).
  4. If the author wants to keep using the dummy variable that takes the value of one if the firm’s financial statement is checked and certified by an external auditor and zero otherwise (financial statement certification), she needs to show a distribution of firms by country and size. Different countries have different accounting requirements for firms of certain sizes, which require them to have external auditors anyway. So many firms will have those by default because their country’s legal system requires it.
  5. In line 152, the author writes “we adopt a database that has been also described in Pietrovito and Pozzolo (2019)”. Please describe the database to the reader of this paper, don’t send the reader to dig into other papers.
  6. In line 260, the author writes “we also control for additional factors that might affect the probability of obtaining a quality certification.” What are the additional factors? Please explain them in detail.
  7. In line 361, the author writes “Financially constrained small firms are therefore 21 per cent less likely to obtain ISO certifications, compared to medium and large ones.” I would interpret that results as 32% less than other small firms that are not financially constraint.
  8. There are a few typos such as foreign line 77, cerditors line 87, Thi footnote 3, etc.

Reviewer 2 Report

Thank you for the opportunity to review this paper. It is clearly written, understands the literature, and attempts to use modern causal inference techniques.

Comment #1

Besides some basic typos and layout issues, the largest problem I have with the paper is that it does not try to justify that is satisfies the exclusion restriction. The authors try to argue by authority that they are following Pietrovito and Pozzolo (2019) in approach. While they are correct in that the their IV strategy is the same, the differences in context make it so in this paper I feel the exclusion restriction is not satisfied. In Pietrovito and Pozzolo (2019) the dependent variable is exports and it is plausible that use of an external auditor is correlated with credit but not with exports. Here, I have a hard time buying that a firm's decision to have its financial statement checked by an external auditor is not also contemporaneously related to ISO-Certifications. This is more problematic in the split sample given that the other instruments are size-based and you then break the sample by firm size.

I understand that the authors have explicitly over-identified the equation so they can do the Sargan-Hansen test. However, the exclusion restriction is not just about a test - that is part of the defense. It is also about theory and the authors have provided no defense of why these instruments are correlated with credit restrictions but not with certification.

Adding to this concern regarding identification, is the fact that the authors do not correctly explain the Sargan-Hansen statistic. Their description states: "overidentifying restrictions statistic (p-value) are obtained from the two-stage least-squaresestimation of the companion specification for the extensive margin of imports, where credit rationing is instrumented using our instruments."  This is puzzling to me, because imports do not appear in this paper. Then I put it into Google Scholar and it appears to come directly from Nucci et al. (2020). So the draft reads as though the authors copied from this paper without realizing what they are doing.

So in addition to a theoretical argument why the authors feel the paper satisfies the exclusion restriction, I would like to see further explicit discussion of the estimation of the Sargan-Hansen test and why the authors fail to reject the null.

Comment #2

The authors should be explicit about how the standard errors in the second stage are corrected given that credit restrictions are estimated.

Comment #3

Given that the authors use a pooled cross-section, I think it would be appropriate to show a sample that only has firms appearing once. If firms are not changing a lot between observations, you are just artificially inflating your standard errors by essentially repeating observations.

Minor comments:

Several typos: Foreign is misspelled as "foreing" at several points. Inappropriate use of the at several points, such as line 79. "...certified companies create more job opportunities than uncertified enterprises."

Round 2

Reviewer 1 Report

The author has tackled serious identification issues that were my main concerns in the last revision. She has addressed all my concerns.

Reviewer 2 Report

Nicely done revision. I would read through the final proof one more time after accepting all the track changes. For example, your Stiglitz citation on page 17 line 706 does not have Economic and Review capitalized.